A draft genome and transcriptome of common milkweed (Asclepias syriaca) as resources for evolutionary, ecological, and molecular studies in milkweeds and Apocynaceae

http://orcid.org/0000-0002-5793-0343 Weitemier Kevin 1 kevin.weitemier@oregonstate.edu
Straub Shannon C.K. 2
http://orcid.org/0000-0003-3099-4387 Fishbein Mark 3
Bailey C. Donovan 4
http://orcid.org/0000-0001-5342-3494 Cronn Richard C. 5
http://orcid.org/0000-0002-3020-6400 Liston Aaron 6
1 Department of Fisheries and Wildlife, Oregon State University , Corvallis, OR , USA
2 Department of Biology, Hobart and William Smith Colleges , Geneva, NY , USA
3 Department of Plant Biology, Ecology, and Evolution, Oklahoma State University , Stillwater, OK , USA
4 Department of Biology, New Mexico State University , Las Cruces, NM , USA
5 Pacific Northwest Research Station, USDA Forest Service , Corvallis, OR , USA
6 Department of Botany & Plant Pathology, Oregon State University , Corvallis, OR , USA
VanBuren Robert
Electronic publication date: 2019 Sep 20
Publication date: 2019
Volume: 7
Electronic Location ID: e7649
Received 2019 Apr 23; Accepted 2019 Aug 9
Copyright year: 2019
License: This is an open access article, free of all copyright, made available under the Creative Commons Public Domain Dedication. This work may be freely reproduced, distributed, transmitted, modified, built upon, or otherwise used by anyone for any lawful purpose.
License URL: https://creativecommons.org/publicdomain/zero/1.0/

Keywords: Asclepias, Milkweed, Apocynaceae, Cardenolide, Chromosome evolution, Genome, Gentianales, Linkage mapping, Plant genome

Funding: National Science Foundation Division of Environmental Biology awards 0919389 and 0919583 Integrative Organismal Systems award 1238731 Funding for this work is provided by the National Science Foundation Division of Environmental Biology awards 0919389 (to Mark Fishbein) and 0919583 (to Richard C. Cronn and Aaron Liston) and Integrative Organismal Systems award 1238731 (to C. Donovan Bailey). The funders had no role in study design, data collection and analysis, decision to publish, or preparation of the manuscript.

==============================
Milkweeds (Asclepias) are used in wide-ranging studies including floral development, pollination biology, plant-insect interactions and co-evolution, secondary metabolite chemistry, and rapid diversification. We present a transcriptome and draft nuclear genome assembly of the common milkweed, Asclepias syriaca. This reconstruction of the nuclear genome is augmented by linkage group information, adding to existing chloroplast and mitochondrial genomic resources for this member of the Apocynaceae subfamily Asclepiadoideae. The genome was sequenced to 80.4× depth and the draft assembly contains 54,266 scaffolds ≥1 kbp, with N50 = 3,415 bp, representing 37% (156.6 Mbp) of the estimated 420 Mbp genome. A total of 14,474 protein-coding genes were identified based on transcript evidence, closely related proteins, and ab initio models, and 95% of genes were annotated. A large proportion of gene space is represented in the assembly, with 96.7% of Asclepias transcripts, 88.4% of transcripts from the related genus Calotropis, and 90.6% of proteins from Coffea mapping to the assembly. Scaffolds covering 75 Mbp of the Asclepias assembly formed 11 linkage groups. Comparisons of these groups with pseudochromosomes in Coffea found that six chromosomes show consistent stability in gene content, while one may have a long history of fragmentation and rearrangement. The progesterone 5β-reductase gene family, a key component of cardenolide production, is likely reduced in Asclepias relative to other Apocynaceae. The genome and transcriptome of common milkweed provide a rich resource for future studies of the ecology and evolution of a charismatic plant family.

Introduction

The development of genomic resources for an ever-increasing portion of the diversity of life is benefiting every field of biology in myriad ways. The decreasing cost of sequencing and the continual development of bioinformatic tools are allowing even single labs and small collaborations to produce genomic content that is beneficial and accessible to the wider research community. This study presents such a resource, including a draft genome assembly of a species in the milkweed genus Asclepias (Apocynaceae).

Asclepias sensu stricto is made up of about 130 species in North and South America (Fishbein et al., 2011). The genus in the Americas is found in a wide range of habitats, from deserts to swamps, plains to shaded forests, and may represent a rapid ecological expansion (Fishbein et al., 2018). The common milkweed, Asclepias syriaca L. (Fig. 1), inhabits wide swaths of eastern North America, westward to Kansas, and northward to Canada (Woodson, 1954). It is well known for the milky latex exuded when injured, showy inflorescences, and pods filled with seeds tufted with fine hairs.

Figure 1 Asclepias syriaca inflorescence.

An inflorescence of A. syriaca. Note the floral coronas (white to light pink) surrounding each central gynostegium. Photo credit: Mark Fishbein.

As members of Apocynaceae subfamily Asclepiadoideae, Asclepias species possess floral architectures unique among plants, including floral coronas and a central gynostegium composed of the unified stamens and pistil (Fig. 1). Most Asclepias species are nearly or entirely self-incompatible (Wyatt & Broyles, 1994), and their pollen is packaged into masses, pollinia, which are transferred as a unit from one flower to another. This usually allows a single successful pollination event to fertilize all of the ovules in an ovary, resulting in full-sibling families in each fruit (Sparrow & Pearson, 1948; Wyatt & Broyles, 1990). The unusual features of Asclepias pollination and floral architecture have positioned it as a model in studies of angiosperm reproductive biology (Broyles & Wyatt, 1990; Wyatt & Broyles, 1990, 1994), floral development (Endress, 2006, 2015), selection on floral characters and prezygotic reproductive isolation (Morgan & Schoen, 1997; La Rosa & Conner, 2017), and floral display evolution (Willson & Rathcke, 1974; Chaplin & Walker, 1982; Fishbein & Venable, 1996).

Milkweeds produce an array of potent secondary compounds, including cardiac glycosides (specifically cardenolides). Some herbivores possess defenses to avoid or tolerate these compounds, including the monarch butterfly, Danaus plexippus. Monarch caterpillars are able to sequester cardenolides from Asclepias to use for their own defense, and Asclepias species are an essential host for monarchs (Brower, Van Brower & Corvino, 1967). The variation within and among Asclepias species in types of and investments in defensive compounds and structures has led to studies of defensive trait evolution (Agrawal & Fishbein, 2006, 2008; Rasmann et al., 2009, 2011; Agrawal et al., 2012; Fishbein et al., 2018; Livshultz et al., 2018), plant-herbivore ecological interactions (Brower, Van Brower & Corvino, 1967; Brower et al., 1972; Vaughan, 1979; Van Zandt & Agrawal, 2004), and plant-herbivore co-evolution (Agrawal & Van Zandt, 2003; Labeyrie & Dobler, 2004; Agrawal, 2005).

A few genomic resources have been developed for Asclepias and other Apocynaceae. The chloroplast and mitochondrial genomes of Asclepias syriaca have been sequenced (Straub et al., 2011, 2013), and flow cytometry estimates place the nuclear genome size of Asclepias syriaca at 420 Mbp (Bainard et al., 2012; Bai et al., 2012). Asclepias is not the first member of Apocynaceae to receive nuclear genome sequencing. Genomic sequencing and assembly of Catharanthus roseus (subfamily Rauvolfioideae) was performed by Kellner et al. (2015) to investigate the production of medicinal compounds (Table 1). Sabir et al. (2016) assembled the genome of Rhazya stricta (Rauvolfioideae) and Hoopes et al. (2018) assembled the Calotropis gigantea (Asclepiadoideae) genome, investigating alkaloid diversity and cardenolide production, respectively (Table 1).

Table 1 Assembly comparison of Asclepias, Calotropis, Catharanthus, Rhazya, and Coffea.

Species	Genome size (Mbp)	Assembly size (Mbp)	N50 (kbp)	# Scaffolds	Sequencing method	
Coffea canephora	710	568.6	1,261	13,345	454 SE & mate-pair, Illumina SE & PE, BACs, haploid accession	
Rhazya stricta	200	274	5,500	980	Illumina PE & mate-pair, PacBio, optical mapping	
Catharanthus roseus	738	506	27.3	41,176	Illumina PE, inbred accession	
Calotropis gigantea	225	157.3	805	1,536	Illumina PE & mate-pair	
Asclepias syriaca	420	156.6	3.4	54,266	Illumina PE & mate-pair	
Notes:

Sequencing method includes technologies and materials used in sequencing.

N50, 50% of the assembly is contained in scaffolds of this length or larger; BAC, bacterial artificial chromosome; SE, single-end; PE, paired-end.

The transcriptomes of several species of Apocynaceae have also been released as part of broader investigations into medicinally important plants, particularly those producing monoterpene indole alkaloids, including Tabernaemontana elegans (Rauvolfioideae), Rauvolfia serpentina (Rauvolfioideae), Rhazya stricta, and Catharanthus roseus (Medicinal Plant Consortium, 2011; Góngora-Castillo et al., 2012a; Xiao et al., 2013; Yates et al., 2014; Park et al., 2014). The transcriptome of Calotropis procera has also been investigated (Kwon et al., 2015; Pandey et al., 2016; Hoopes et al., 2018).

Outside of Apocynaceae the most closely related species to milkweed with a sequenced genome is the diploid ancestor of coffee, Coffea canephora (Rubiaceae; Denoeud et al., 2014). Coffea is in the same order as Asclepias, Gentianales, and Coffea canephora has the same number of chromosomes: x = n = 11, 2n = 22 (Denoeud et al., 2014). The Coffea genome assembly is a high-quality reference, with large scaffolds ordered onto pseudochromosomes (scaffolds that have been ordered based on linkage information, as though on a chromosome; Table 1).

The genomic assembly of Asclepias syriaca presented here includes a nearly complete representation of gene space, supported by transcriptome evidence. The heterozygosity present in this obligate outcrossing species is used to develop a panel of single nucleotide polymorphisms (SNPs) that can be captured via targeted enrichment, and a set of offspring from the sequenced individual is used to cluster assembled scaffolds into linkage groups. A comparison of linkage groups between Asclepias and Coffea is presented, providing insights into chromosome organization in Asclepias, and chromosomal evolution within Gentianales. Both genome and transcriptome sequences are used to explore gene family evolution, especially as related to cardenolide biosynthesis.

Methods

Tissue preparation and library construction

Leaf tissue of Asclepias syriaca was sampled from a single individual at the Western Illinois University research farm, raised from seed from a wild population in McDonough County, Illinois (40.29622°N, 90.89876°W; Winthrop B. Phippen s.n., OSC 226164, 226165). DNA was extracted from frozen tissue using the FastDNA Spin Kit from MPBiomedicals (Santa Ana, CA, USA) following manufacturer’s protocols, modified by the addition of 40 μl 1% polyvinylpyrrolidone and 10 μl β-mercaptoethanol to the 1,000 μl lysis solution (800 μl CLS-VF + 200 μl PPS) prior to grinding.

Aliquots of isolated DNA were sheared with a BioRuptor sonicator (Diagenode Inc., Denville, NJ, USA) at low power for 10 cycles of 30 s on/30 s off. Two libraries were prepared using the Illumina Paired-End DNA Sample Prep Kit (catalog number PE-102-1001; Illumina Inc., San Diego, CA, USA; Solexa, Inc, 2006). Ligated fragments were cut from agarose gels centered around 225 and 450 bp, and were amplified through 15 and 14 cycles, respectively, of polymerase chain reaction using Phusion High-Fidelity PCR Master Mix (New England BioLabs, Ipswich, MA, USA) and standard Illumina primers. Cleaned product was submitted for sequencing on an Illumina GAII Sequencer at the Center for Genome Research and Biocomputing (CGRB) at Oregon State University (Corvallis, OR, USA). One lane of the 450 bp library was sequenced with 80 bp paired-end reads, and five lanes of the 225 bp library were sequenced with 120 bp paired-end reads.

Frozen tissue of the sequenced individual was sent to GlobalBiologics, LLC (Columbia, MO, USA) for DNA extraction and production of mate-pair libraries using the Illumina Mate Pair Library v2 protocol with average insert sizes of 2,750 and 3,500 bp, and indexed with unique barcode sequences (Bioo Scientific, Austin, TX, USA). The 2,750 bp library was sequenced with 101 bp paired-end reads on an Illumina HiSeq 2000 sequencer at the CGRB, on the same lane as two other samples from unrelated projects. The 3,500 bp library was sequenced on an Illumina MiSeq at Oregon Health and Science University (Portland, OR, USA) with 33 bp paired-end reads (Table 2). Purified DNA of that same individual was also provided to the CGRB for production of a mate-pair library using the Illumina Nextera protocol with an average insert size of 2,000 bp. This library was sequenced with 76 bp paired-end reads on an Illumina MiSeq at the CGRB, along with 14 other samples from unrelated projects (Table 2).

Table 2 Asclepias syriaca sequencing summary.

Library type	Insert size (bp)	Machine	Lanes	Read length (bp)	Clusters	Raw yield (Mbp)	Processed yield (Mbp)	SRA	
Paired-end	225	GA II	5	120	193,332,028	46,400	29,171	SRX2164079	
Paired-end	450	GA II	1	80	22,244,539	3,559	1,530	SRX322144	
Mate-pair	2,000	MiSeq	1/15	76	257,750	39	34	SRX2164126	
Mate-pair	2,750	HiSeq 2000	1/3	101	46,704,483	9,434	2,819	SRX322145	
Mate-pair	3,500	MiSeq	1	33	5,815,961	384	195	SRX322148	
RNA-Seq Buds	–	HiSeq 2000	1/4	101	48,085,747	4,857	2,812	SRX2432900	
RNA-Seq Leaf	–	HiSeq 2000	1/4	101	64,772,831	6,542	3,787	SRX2435668	
Paired-end total	215,576,567	49,959	30,701		
Mate-pair total	52,778,194	9,857	3,048		
RNA-Seq total	112,858,578	11,399	6,599		
Note:

Machine, Illumina instrument that performed the sequencing; Raw yield, Processed yield, Total Mbp of sequence data before and after read processing. SRA, NCBI Short Read Archive accession number.

Genomic read processing

Read pools were evaluated for quality parameters using FastQC (Andrews, 2010). Pairs of reads properly mapping to the Asclepias chloroplast or mitochondria, with three or fewer mismatches between the target and query, were filtered out using Bowtie 2 v. 2.1.0 (scoring parameter “--score-min L,-6,0”), samtools v. 0.1.18, and bamtools v. 2.3.0 (Li et al., 2009; Langmead & Salzberg, 2012; Barnett et al., 2013). Portions of reads matching the Illumina adapter sequences were removed with Trimmomatic v. 0.30 and the “ILLUMINACLIP:TruSeq2-PE.fa:2:30:10” option (Bolger, Lohse & Usadel, 2014). Duplicate read pairs from the same library were removed using the custom script fastq_collapse.py (Weitemier, 2014). Paired-end read pairs with sequences that overlapped by ≥7 bp sharing ≥90% identity were merged using the program FLASH v. 1.2.6 (parameters “-m 7 -M 80 -x 0.10”) (Magoč & Salzberg, 2011). The 3′ and 5′ ends of reads were then trimmed of any bases with a Phred quality score below 30, and any remaining reads less than 30 bp were removed using Trimmomatic command “LEADING:30 TRAILING:30 MINLEN:30.”

Summary statistics were calculated using a k-mer distribution plot of reads from the 225 bp insert library after removing chloroplast and mitochondrial reads, but prior to joining with FLASH. K-mers of 17 bp were counted using BBTools script kmercountexact.sh, and estimates of genome size and heterozygosity were calculated using the program gce (Liu et al., 2013; Bushnell & Rood, 2015).

RNA-seq library preparation, sequencing, and assembly

Total RNA was extracted from the individual used for genome sequencing from leaves and buds separately, by homogenizing approximately 200 mg of fresh frozen tissue on dry ice in a Fast-Prep-24 bead mill. Cold extraction buffer (1.5 ml of 3M LiCl/8M urea; 1% PVP K-60; 0.1M dithiothreitol; Tai, Pelletier & Beardmore, 2004) was added to the ground tissue. Tissue was then homogenized and cellular debris pelleted at 200×g for 10 min at 4 °C. Supernatant was incubated at 4 °C overnight. RNA was pelleted by centrifugation (20,000×g for 30 min at 4 °C) and cleaned using a ZR Plant RNA MiniPrep kit (Zymo Research, Irvine, CA, USA). The integrity of the extracted RNA was assessed using an Agilent 2100 Bioanalyzer (Agilent Technologies, Santa Clara, CA, USA); extractions from both tissues showed RIN values greater than 8.0. For each tissue type, an RNA-seq library was prepared using the Illumina RNA-Seq TruSeq kit v. 2.0 with the modifications of Parkhomchuk et al. (2009) to allow strand-specific sequencing by dUTP incorporation.

Libraries were sequenced on an Illumina HiSeq 2000 at the CGRB to yield 101 bp single-end reads. Before further analysis, reads that did not pass the Illumina chastity and purity filters were removed. Trimmomatic 0.20 (Bolger, Lohse & Usadel, 2014) was used to trim the final base of each read, leading and trailing bases with quality scores below Q20, and all following bases if a sliding window of five bp did not have an average quality of at least Q30. Reads shorter than 36 bp after trimming were excluded (Trimmomatic command “CROP:100 LEADING:20 TRAILING:20 SLIDINGWINDOW:5:30 MINLEN:36”).

Transcripts were assembled de novo using Trinity (Release 2013-08-14) (Grabherr et al., 2011) for bud and leaf reads separately, as well as combined into a single data set using default settings, except for using a minimum contig length of 101 bp. The same settings were also used to assemble RNA-seq data from leaf tissue of the same Asclepias syriaca individual from a library made using ribosomal RNA subtraction (Straub et al., 2013). Best-scoring open reading frames (ORFs) were determined for each library based on attributes including length, reading frame, and nucleotide composition using the TransDecoder utility provided with Trinity (Haas et al., 2013). Transcripts were annotated using Mercator (Lohse et al., 2014) and TRAPID (Van Bel et al., 2013). The Mercator analysis was conducted with default options, with the exception of not allowing multiple bin assignments. Therefore multiple databases, including UniProt, were used in annotation. The TRAPID annotations were based on the Plaza 2.5 reference database (Van Bel et al., 2012), and the similarity search was restricted to the eudicot clade with an E-value cutoff of 10e−5. Functional annotations were added to transcripts based on both gene family and best database hit.

Comparative transcriptome and gene family evolution analyses in Apocynaceae

For a comparative analysis, transcriptomes were obtained for five other species of Apocynaceae. Catharanthus roseus and Rauvolfia serpentina transcriptomes were downloaded from the Medicinal Plant Genomics Resource project database (http://medicinalplantgenomics.msu.edu; Góngora-Castillo et al., 2012b), the Rhazya stricta (GenBank GAMW01000000; Yates et al., 2014) and Calotropis procera (GenBank GBHG01000000; Kwon et al., 2015) transcriptomes were downloaded from National Center for Biotechnology Information (NCBI), and the Tabernaemontana elegans transcriptome was downloaded from the PhytoMetaSyn Project database (https://bioinformatics.tugraz.at/phytometasyn; Xiao et al., 2013). All transcriptomes, including that of Asclepias syriaca, were checked for duplicate transcripts, and the duplicates removed using the Dedupe tool in BBMap (Bushnell & Rood, 2015). Transcriptomes were checked for completeness using BUSCO v. 1.22 (Simão et al., 2015). Transcripts of all species were assigned to reference gene families using TRAPID. Reference gene family assignments were obtained from two high quality genomes, Coffea canephora (Denoeud et al., 2014) and Vitis vinifera (PLAZA v. 2.5; Proost et al., 2009).

A phylogenetic framework for comparative analysis was produced using published evolutionary relationships and divergence times in Apocynaceae (Fishbein et al., 2018). The timings of the Coffea split from Apocynaceae and the Vitis split from Gentianales were based on the estimates of Wikström et al. (2015). In order to examine changes in gene family sizes across Apocynaceae transcriptomes, BadiRate v. 1.35 (Librado, Vieira & Rozas, 2012) was run using the birth-death-innovation stochastic model with a free rate branch model where each branch can have a different gene turn-over rate. Gains and losses were inferred using Wagner (ordered) parsimony (Kluge & Farris, 1969).

Genomic sequence assembly

Processed read-pairs were assembled into contigs using Platanus v. 1.2.1 (Kajitani et al., 2014). Platanus is designed to assemble highly heterozygous diploid genomes, and initially uses several k-mer sizes during assembly. Asclepias reads were assembled with an initial k-mer size of 25 bp with a k-mer step increase of 10 bp up to a maximum k-mer of 110 bp. As part of the expectation for heterozygous assembly, Platanus can merge contigs sharing high identity. We allowed contigs sharing 85% identity to be merged (assembly parameters “-k 25 -u 0.15”).

Scaffolding was performed with Platanus, setting the paired-end reads as “inward pointing” reads and the mate-pair reads as “outward pointing” reads. Reads were mapped to scaffolds using an initial seed size of 21 bp, one link between contigs was sufficient to align them into a scaffold, and scaffolds sharing 85% identity could be merged (scaffolding parameters “-s 21 -l 1 -u 0.15”).

Gaps between scaffolds were closed via local alignment and assembly of reads around the gaps using Platanus. An initial seed size of 21 bp was used to include reads in the mapping around a gap, and a minimum overlap of 21 bp between the newly assembled filler contig and the edges of the scaffold was required to use that contig to fill the gap (gap close parameters “-s 21 -k 21 -vd 21 -vo 21”).

Transcripts were mapped to Asclepias scaffolds ≥1 kbp using BLAT v. 32×1; one or more transcripts spanning multiple scaffolds were used to merge those scaffolds (Kent, 2002). This was performed with the program Scubat (https://github.com/elswob/SCUBAT; accessed December 17, 2015) modified so that scaffolds would not be clipped when joined by cap3 v. 02/10/15 (Huang & Madan, 1999; Tange, 2011; Elsworth, 2012).

Contaminant removal

Merged scaffolds were compared against a genomic database of potentially contaminating organisms with the program DeconSeq standalone v. 0.4.3 (Schmieder & Edwards, 2011). Contaminant databases were downloaded from the DeconSeq website representing bacteria, archaea, viruses, 18S rRNA, zebrafish, mouse, and several human genomes (http://deconseq.sourceforge.net; accessed January 20, 2016). Fungal genomes were obtained from the NCBI including Alternaria arborescens accession AIIC01, Aspergillus fumigatus AAHF01, Bipolaris maydis AIHU01, Botrytis cinerea assembly GCA_000832945.1, Cladosprium sphaerospermum AIIA02, Fomitopsis pinicola AEHC02, Fusarium oxysporum AAXH01, Galerina marginata AYUM01, Hypoxylon sp. JYCQ01, Penicillium expansum AYHP01, Rhodotorula graminis JTAO01, Saccharomyces cerevisiae assembly GCA_000146045.2, and Trichoderma reesei AAIL02 (Goffeau et al., 1997; Nierman et al., 2005; Martinez et al., 2008; Ma et al., 2010; Amselem et al., 2011; Hu et al., 2012; Ohm et al., 2012; Ng et al., 2012; Floudas et al., 2012; Riley et al., 2014; Firrincieli et al., 2015; Shaw et al., 2015; Li et al., 2015). The genome of Solanum lycopersicum (ITAG 2.4) was downloaded from the Sol Genomics Network (The Tomato Genome Consortium, 2012). The fungal and Solanum genomes were prepared as DeconSeq databases following the DeconSeq website, including filtering of repeated Ns, removal of duplicate sequences, and indexing with a custom version of BWA released with DeconSeq (Li & Durbin, 2010; http://deconseq.sourceforge.net; accessed January 20, 2016).

Genomes obtained from the DeconSeq website and the fungal genomes were used as contaminant databases, the Solanum genome was used as a retain database. Scaffolds matching one of the contaminant genomes with ≥80% identity along ≥80% of the scaffold length were excluded as contaminants. Those scaffolds matching both a contaminating genome and the Solanum genome were retained.

Gene prediction and annotation

A library of Asclepias repetitive elements was created following guidelines in the MAKER Genome Annotation Pipeline online documentation (Jiang, 2015). The program RepeatModeler v. open-1.0.8 was used to integrate the programs RepeatMasker v. open-4.0.5, rmblastn v. 2.2.28, RECON v. 1.08, Tandem Repeats Finder v. 4.07b, and RepeatScout v. 1.0.5 (Benson, 1999; Bao & Eddy, 2002; Price, Jones & Pevzner, 2005; Smit, Hubley & Green, 2015). Repeat models initially missing a repeat annotation were compared, using BLAT, against a library of class I and class II transposable elements acquired from the TESeeker website (Kennedy et al., 2010, 2011), and matching sequences provided an annotation. Remaining unannotated models were submitted to the online repeat analysis tool, CENSOR, and provided annotations with a score ≥400% and ≥50% sequence similarity (Kohany et al., 2006). A set of proteins from Arabidopsis thaliana was filtered to remove proteins from transposable elements, then compared using BLASTX against the Asclepias repeat models. The program ProtExcluder.pl v. 1.1 then used the BLASTX output to remove repeat models and flanking regions matching Arabidopsis proteins (Altschul et al., 1990; Jiang, 2015).

The set of scaffolds ≥1 kbp were annotated via the online annotation and curation tool GenSAS v. 4.0 (Lee et al., 2011; Humann et al., 2016), which was used to implement the following tools for repeat masking, transcript and protein mapping, ab initio gene prediction, gene consensus creation, and mapping of Asclepias predicted proteins:Repeats in the assembled sequence were masked via RepeatMasker v. open-4.0.1 using the Asclepias repeat models and using models developed from dicots more broadly (Smit, Hubley & Green, 2015).

Multiple datasets were mapped onto Asclepias scaffolds in order to assist with gene prediction. The best-scoring ORFs from assembled Asclepias transcripts were mapped using both BLAT and BLAST (expect <1e-50, 99% identity). Assembled transcripts from Calotropis procera were mapped with BLAT (Kwon et al., 2015). Proteins from Coffea canephora were mapped with BLASTX (e < 0.0001; Denoeud et al., 2014). While additional high-quality genomes within Apocynaceae were later released (Sabir et al., 2016; Hoopes et al., 2018), they were not available at the time this work was performed.

Genes were predicted using the ab initio tools Augustus v. 3.1.0, SNAP, and PASA (Haas et al., 2003; Korf, 2004; Stanke et al., 2008). Augustus was run using gene models from Solanum, finding genes on both strands, and allowing partial models; SNAP was run using models from Arabidopsis thaliana. PASA was informed by the best-scoring ORFs from assembled Asclepias transcripts.

Multiple lines of evidence were integrated into a gene consensus using EVidenceModeler (Haas et al., 2008) with the following weights: Augustus, 1; SNAP, 1; Coffea proteins, 5; Asclepias transcripts (BLAST), 7; Asclepias transcripts (BLAT), 7; Calotropis transcripts, 5; PASA, 7. Consensus gene models were then refined using PASA, again informed by Asclepias transcripts.

Predicted proteins were compared to the NCBI plant RefSeq database using BLASTP (expect <1e-4, BLOSUM62 matrix; Pruitt et al., 2002), as well as being mapped against protein sequences from Coffea and Catharanthus roseus (expect <1e-4; Denoeud et al., 2014; Kellner et al., 2015). Protein families were classified using the InterPro database and InterProScan v. 5.8-49.0 (Jones et al., 2014; Mitchell et al., 2015). Transfer RNAs were identified using tRNAscan-SE v. 1.3.1 (Lowe & Eddy, 1997). Additional ORFs were found using the getorf tool from the EMBOSS suite, accepting a minimum of 30 bp (Rice, Longden & Bleasby, 2000).

Some predicted proteins were missing one or more exons, either because they were fragmented on the ends of scaffolds or, rarely, transcript evidence predicted exons with non-canonical splice sites. The predicted coding sequence produced by GenSas for some of these proteins was out of frame. In these cases the coding sequence was translated under all reading frames and a translation lacking internal stop codons was selected, if available.

An estimate of the completeness of the assembled gene space was calculated using the program BUSCO v. 1.22 and a set of 956 conserved single copy plant genes (Simão et al., 2015). BUSCO was run independently on the set of coding sequences returned following gene prediction as well as on the assembled scaffolds ≥1 kbp using Augustus gene prediction with Solanum models. Predicted genes from Asclepias, Catharanthus, Coffea, and Vitis (obtained from the PLAZA 3.0 database) were clustered into orthogroups using OrthoFinder v. 0.7.1 (The French-Italian Public Consortium for Grapevine Genome Characterization, 2007; Emms & Kelly, 2015; Proost et al., 2015).

Gene analyses

The progesterone 5β-reductase (P5βR) region (PLAZA v. 2.5 gene family HOM000752; InterPro NAD(P)-binding domain IPR016040; Gene Ontology: coenzyme binding GO:0050662, catalytic activity GO:0003824) was identified in assembled scaffolds with BLAT (Kent, 2002), using the P5βR sequences from Asclepias curassavica (ADG56538; Bauer et al., 2010) and Catharanthus roseus (KJ873882–KJ873887; Munkert et al., 2015) as references. A maximum likelihood tree was constructed from peptide sequences of two Asclepias syriaca regions with high identity to P5βR; six Catharanthus P5βR sequences; the Asclepias curassavica sequence; P5βR sequences from Calotropis procera (Kwon et al., 2015), Calotropis gigantea (Hoopes et al., 2018), and Rhazya stricta (Sabir et al., 2016); sequences from Digitalis purpurea and Digitalis lantata (ACZ66261, AAS76634), representing P5βR2 and P5βR paralogs, respectively; and a sequence from Picea sitchensis (ABK24388). P5βR sequence alignments were performed using MUSCLE 3.8.425, as implemented in Geneious v. 11.1.5, with a maximum of 10 iterations (Edgar, 2004; Kearse et al., 2012). The optimal models of amino acid substitution, rate variation among sites, and equilibrium frequencies were inferred using the Akaike and Bayesian information criteria, as implemented in the online tool PhyML 3.0, which was also used to infer trees under those models and calculate aBayes support values (Guindon & Gascuel, 2003; Guindon et al., 2010; Anisimova et al., 2011).

SNP finding and targeted enrichment probe development

The Platanus genome assembler uses a de Bruijn graph approach for contig assembly (Kajitani et al., 2014). Certain types of branches in this graph, known as “bubbles,” may be caused by heterozygosity and are saved by the program for use in later assembly stages. Here, saved bubbles were filtered to identify those likely to represent heterozygous sites in low-copy regions of the genome.

The program CD-HIT-EST v. 4.5.4 was used to cluster any bubbles sharing ≥90% identity, which were removed, leaving only unique bubbles (Li & Godzik, 2006). Unique bubbles were mapped against the set of Asclepias scaffolds ≥1 kbp using BLAT at minimum identity thresholds of 90% and 95% (Kent, 2002). A set of 4,000 SNP probes developed from a preliminary study using a similar approach, but from a different genome assembly, were mapped against the assembly presented here with a 90% identity threshold (Weitemier et al., 2014). One appropriate bubble from each scaffold <10 kbp, and up to two bubbles from scaffolds ≥10 kbp, were selected, up to a total of 20,000 bubbles. Bubbles mapping only once within the ≥90% identity mapping analysis were selected first, progressively adding bubbles that either mapped to ≤4 locations in the ≥90% identity mapping or mapped to ≤3 locations in the ≥95% identity mapping. Bubble sequences were trimmed to 80 bp, and centered around the SNP site where possible. Potential SNP probes were further analyzed by MYcroarray (now Arbor BioSciences, Ann Arbor, MI, USA) and excluded if they were predicted to anneal in a solution hybridization reaction to >10 locations within the Asclepias genome at 62.5–65 °C or >2 locations above 65 °C. A total of 20,000 RNA oligos suitable for targeted enrichment, matching 17,684 scaffolds, were produced by MYcroarray. RNA oligo sequences are available in the supplemental data set (Weitemier, 2017).

Linkage mapping population

Mature follicles were collected from the open-pollinated plant that was the subject of genome sequencing. Approximately 100 seeds from six follicles collected from four stems of this plant (1, 3, 1, and 1 follicle per stem) were germinated and grown at Oklahoma State University. Due to the pollination system of Asclepias, seeds in a fruit are likely to be fertilized by a single pollen donor (Sparrow & Pearson, 1948; Wyatt & Broyles, 1990), meaning up to six paternal parents are represented among the 96 mapping offspring.

Seeds were surface sterilized in 5% bleach and soaked for 24 h in distilled water. The testa was nicked opposite from the micropylar end and the seeds germinated on moist filter paper, in petri dishes, in the dark, at room temperature. Germination occurred within 4–7 days, and seedlings were planted into MetroMix 902 media in plug trays when radicles attained a length of two to three cm. Seedlings were again transplanted to three-inch deep pots following the expansion to two sets of true leaves. Seedlings were grown under high intensity fluorescent lights in a controlled environment chamber at 14 h daylength at approximately 27 °C. Plants were grown for approximately 90 days, harvested, and rinsed in distilled water, and frozen at −80 °C. DNA was extracted from roots, shoots, or a combination of roots and shoots using the FastDNA® kit (MP Biomedicals, Santa Ana, California) and Thermo Savant FastPrep® FP120 Cell Disrupter (Thermo Scientific, Waltham, MA, USA). DNA quantity and quality were visualized using agarose gel electrophoresis and quantified with a Qubit® fluorometer (Invitrogen, Carlsbad, CA, USA) and Quant-iT™ DNA-BR Assay Kit.

A total of 96 genomic DNA samples were diluted as necessary with ultrapure water to obtain approximately three μg in 100 μl and sheared on a Bioruptor UCD-200 (Diagenode Inc., Denville, NJ, USA) at low power for 12 cycles of 30 s on/30 s off. Several samples required sonication for 5–10 additional cycles to achieve a high concentration of fragments at the target size of 300–400 bp. Illumina-compatible, dual-indexed libraries were produced with the TruSeq® HT kit (Illumina, San Diego, CA, USA), each with a unique barcode.

Barcoded libraries were pooled by equal DNA mass in three groups of 32 samples. These were enriched for targeted SNP regions using RNA oligos and following MYcroarray MYbaits protocol v. 3.00. Enriched pools were then themselves evenly pooled and sequenced with 150 bp paired-end reads on an Illumina HiSeq 3000 at the CGRB, producing 120.3 Mbp of sequence data (NCBI short read archive: SRX2163716–SRX2163811).

Linkage analyses

Reads from the 96 target-enriched offspring libraries were processed using Trimmomatic v. 0.33 to remove adapter sequences, bases on the ends of reads with a Phred quality score below three, and clipping once a sliding window of four bp fell below an average quality score of 17 (Bolger, Lohse & Usadel, 2014). Processed reads for 90 samples (excluding six with low sequencing depth) were mapped onto the assembled scaffolds using bowtie2 with “sensitive” settings and a maximum fragment length of 600 bp (Langmead & Salzberg, 2012). Reads from the 225 bp insert library of the sequenced individual were also mapped back onto assembled scaffolds using the same settings. Mappings for all individuals and the parent were combined using samtools v. 0.1.16 with the samtools “mpileup” command and flags “-D -S” to record the per-sample read depth and strand-specific bias. SNP positions were called using the bcftools “view” command with flags “-v -c -g” to output only potential variant sites with called genotypes (Li et al., 2009).

Two subsets of SNPs were retained. The first was a subset of SNPs where the maternal parent was heterozygous and the paternal parents for all offspring were homozygous for the same allele. The file containing all variants was converted to a format suitable for the R package OneMap, using a custom perl script (Tennessen, 2015), retaining only sites heterozygous in one parent, the maternal sequenced individual. In this filtering the minor genotype abundance (either heterozygote or homozygote) needed to be at least 24 across 90 samples, loci could have up to 30% missing individuals, and alternative genotypes within individuals were ignored if their Phred probability score was 15 or above (i.e., of the three possible genotypes AA, Aa, aa, one should be most probable with a low Phred score and the other two less probable with Phred scores above 15).

The second subset retained SNPs from 22 full siblings (from the fruit producing the most offspring) for loci in which either the maternal or paternal parent, but not both, were heterozygous. Filtering in this set required a minor genotype abundance of at least five, loci could have up to four missing individuals, and genotypes with Phred probabilities of 20 or above were ignored (i.e., the final genotype calls are more certain because alternative genotypes are less likely).

SNP sets were clustered into linkage groups in R v. 3.2.2 using the package OneMap v. 2.0-4 (Margarido, Souza & Garcia, 2007; R Core Team, 2014). One SNP from each scaffold was selected from SNPs among the full set of individuals, and were grouped using a logarithm of odds (LOD) threshold of 8.4. This clustered SNP loci into 11 clear groups, referred to here as the core linkage groups.

From the full-sibling set of SNPs, those held on the same scaffold and with identical genotypes across individuals (i.e., in perfect linkage) were grouped, and SNPs on different scaffolds in perfect linkage with no missing data were grouped. This was performed separately for loci where either the maternal or paternal parent was heterozygous. These loci were clustered into groups using LOD scores 6.1, 6.0, and 5.5. Each of these groupings produced hundreds of groups, but each contained about 22 groups that were substantially larger than the others.

A custom R script was used to combine the linkage group identity of scaffolds in the core linkage groups with scaffolds and groups in the sibling sets (Weitemier, 2017). For example, scaffold A could be assigned to a linkage group if it was in perfect linkage in the sibling set with scaffold B, and scaffold B was also present in the core linkage groups. If multiple scaffolds were perfectly linked, but associated with different core linkage groups, no unknown scaffolds would be assigned unless the most common core linkage group was three times as common as the next core group.

Linkage groupings in the sibling sets could be assigned to core linkage groups based on the membership of the scaffolds they contained. If the markers indicating that a sibling group should belong to a certain core linkage group were 10 times as common as markers supporting a second most common assignment, then the sibling group was assigned to the core group, and all unknown scaffolds it contained also assigned to that group. (For example, sibling group A contains 10 scaffolds known to be on core linkage group 1, one scaffold known to be on core linkage group 2, and one unknown scaffold; sibling group A is assigned to core linkage group 1 and the unknown scaffold is similarly assigned.)

This process was performed iteratively, progressively assigning scaffolds to core linkage groups. It was performed first with the sibling set grouped with LOD 6.1, then the grouping with LOD 6.0, finally the grouping with LOD 5.5.

Synteny within Gentianales

Scaffolds found within the core linkage groups were mapped to Coffea coding sequences (BLASTN, expect <1, best hit chosen) and mapped to their location on Coffea pseudochromosomes. Six Asclepias linkage groups had a roughly one-to-one correspondence with a Coffea pseudochromosome (e.g., most of the scaffolds from that linkage group, and few from other linkage groups, mapped to the pseudochromosome). From these six linkage groups one marker was selected for every one Mbp segment of the Coffea chromosome. Recombination fractions were measured among these loci using OneMap (retaining “safe” markers with THRES=5) and converted to cM using the Kosambi mapping function.

Results

Sequencing and read processing

Paired-end sequencing produced 215.6 million pairs of reads representing 50.0 Gbp of sequence data, and mate-pair sequencing produced 52.8 million pairs of reads for 9.9 Gbp of sequence data. After read filtering and processing, 30.7 Gbp of paired-end sequence data remained along with 3.0 Gbp of mate-pair data. This represents total average sequence coverage of 80.4× on the 420 Mbp Asclepias syriaca genome (Table 2).

The distribution of 17 bp k-mers from the largest set of paired-end reads demonstrates a clear bi-modal distribution, with peaks at 43× and 84× depth (Fig. S1), corresponding to the sequencing depth of heterozygous and homozygous portions of the genome, respectively. This k-mer distribution provides a genome size estimate of 406 Mbp, and a site heterozygosity rate estimate of 0.056.

Sequence assembly and gene annotation

The assembly of Asclepias syriaca contains 54,266 scaffolds ≥1 kbp, with N50 = 3,415 bp, representing 37% of the estimated genome (156.6 Mbp of sequence plus 5.8 Mbp of gaps, Table 3). When including scaffolds ≥200 bp the assembly sums to 229.7 Mbp, with N50 = 1,904 bp. The largest scaffold is 100 kbp, and 10% of the Asclepias genome, 42.82 Mbp, is held on 2,343 scaffolds ≥10 kbp. Prior to scaffolding, gap closing, contaminant removal, and transcript-assisted scaffolding, the initial assembly produced 2.8 million contigs, with 848,509 ≥ 200 bp and 38,615 ≥ 1 kbp. Initial contigs ≥1 kbp summed to 74.9 Mbp with N50 = 2,041 bp, and the largest contig was 16 kbp.

Table 3 Asclepias syriaca assembly statistics.

Minimum scaffold	Sum (Mbp)	N80	N50	N20	# scaffolds	
77 (all)	265.9	317	1,454	7,080	508,851	
200	229.7	621	1,904	8,967	221,940	
1,000	156.6	1,633	3,415	14,019	54,266	
10,000	42.82	12,894	18,998	30,689	2,343	
Note:

Minimum scaffold: The minimum scaffold size (bp) used for calculations. Sum: The sum of the lengths of all included scaffolds, not including gaps. N80, N50, N20: The length (bp) of the shortest scaffold in the set of largest scaffolds needed to equal or exceed (N/100)(Sum). # scaffolds: Total scaffolds ≥ the minimum size.

Within the 156.6 Mbp of scaffolds ≥1 kbp, 1.25 million putative ORFs were identified, along with 193 transfer RNA loci. Assembled repeat elements made up about 75.7 Mbp. A total of 14,474 protein-coding genes were identified based on transcript evidence, closely related proteins, and ab initio models. These are predicted to produce 15,628 unique mRNAs, and are made up of a total of 87,496 exons with an average length of 225.3 bp. The median length of predicted proteins is 303 amino acids (mean = 402), which is shorter than lengths predicted in Calotropis (median = 367, mean = 448), similar to those predicted in Coffea (median = 334, mean = 402), but longer than those predicted in Catharanthus (median = 251, mean = 340; Fig. 2). Of the 14,474 predicted genes, 13,749 (95.0%) mapped to either Coffea or Catharanthus proteins, and 9,811 mapped to RefSeq proteins.

Figure 2 Peptide length histograms of Asclepias, Calotropis, Coffea, and Catharanthus.

Mean and median peptide lengths are provided in the legend.

Assembly of the Asclepias transcriptome produced 32,728 best-scoring ORFs, ranging from 300 to 13,005 bp, with N50 = 1,422 bp, and summing to 37.4 Mbp. Of these, 31,654 (96.7%) mapped onto scaffolds ≥1 kbp. For Calotropis, 92,115 (88.4%) transcripts were mapped to Asclepias scaffolds, while 23,182 (90.6%) proteins from Coffea mapped to the assembly. BUSCO analysis of the genome assembly identified 895 of the 956 plant genes in its set (93.6%). Of these, 209 were identified as duplicates and another 77 genes were fragmented, meaning they were found in the genome assembly, but with a length outside two standard deviations of the mean BUSCO length for that gene. When applied to just the set of coding sequences BUSCO identified 742 complete genes (302 duplicated) and 84 fragmented genes, representing 86.4% of the conserved plant gene set. Apocynaceae transcriptomes were compared using the BUSCO set of 429 genes common to eukaryotes. The Asclepias transcriptome contained 365 of the genes (117 duplicated, 21 fragmented), representing 85.1%. Presence of these genes in other transcriptomes (Catharanthus, Rauvolfia, Rhazya, Tabernaemontana, Calotropis) ranged from 83.7% in Calotropis to 86.5% in Tabernaemontana, indicating that the Asclepias transcriptome assembly was of similar completeness to Apocynaceae transcriptomes publically available at the time of analysis. All Apocynaceae transcriptomes showed increased duplication of the 429 genes with approximately 2× the number of duplicates on average compared to the Coffea, Catharanthus, and Vitis genomes.

Among 100,114 predicted genes from Asclepias, Catharanthus, Coffea, and Vitis, 69.9% were clustered into 13,906 orthogroups. Asclepias had the highest percentage of genes placed in orthogroups, 81.6%, but those genes only represent 9,837 orthogroups, the lowest of the four genomes. Asclepias shared the fewest orthogroups with other species (Table S1).

Comparison of all six Apocynaceae transcriptomes showed 5,195 gene families were common to all. The Asclepias transcriptome contained 5,762 gene families also present in the Coffea genome. There were 58 gene families with 1−3 gene copies in Asclepias that were not present in other Apocynaceae. Among Apocynaceae lineages, Asclepias was not unusual in its number of gene gains or losses based on the BadiRate analysis. Asclepias had close to the median number of gene gains among all lineages with 5,697 (median = 5,791.5), much less than the 15,831 gene gains inferred in the lineage with the highest number of gains, Rauvolfia. Similarly, the number of gene losses in Asclepias at 905 was just below the median number of losses (median = 1,136), and much less than the 7,619 losses inferred for Catharanthus. While Asclepias had one of the highest gene birth rates over time (0.01082 events per gene per million years; Fig. 3), it was lower than that of close relative Calotropis (0.01463 events per gene per million years), and the rate inferred for the Rauvolfia plus Catharanthus plus Tabernaemontana lineage (0.14406 events per gene per million years) was an order of magnitude greater. Asclepias had close to the median value for gene death rate (0.00177 events per gene per million years). However, Asclepias had the second highest gene innovation rate (0.00069 events per gene per million years) compared to other lineages (Fig. 3). As with gene birth rate, the gene innovation rate of the Rauvolfia plus Catharanthus plus Tabernaemontana lineage (0.00146 events per gene per million years) was an order of magnitude higher.

Figure 3 Gene family evolution in Apocynaceae inferred from transcriptomes.

The ultrametric tree depicts the phylogenetic relationships and estimated divergence times of sampled Apocynaceae and outgroups (Coffea, Vitis). The number of gene birth/death/innovation events per gene per million years across all gene families is shown above the branches. Numbers following tip labels represent the observed number of P5βR gene family paralogs, and the inferred number of paralogs present in common ancestors is shown to the right of nodes.

Linkage mapping and synteny within Gentianales

Following filtering, the set of all 96 offspring retained over 16,000 SNPs for which the maternal parent was heterozygous and all the paternal parents were homozygous for the same allele. These were located on 8,495 scaffolds, covering 43.5 Mbp. A total of 90 of 96 individuals were sequenced at adequate depth to inform linkage group analyses. At a logarithm of odds (LOD) score of 8.4, 7,809 scaffolds were clustered into 11 groups, the core linkage groups, representing 41.9 Mbp.

Filtering for SNPs among just the largest group of full-siblings, in which one parent (but not both) was heterozygous, found 83,854 SNPs on 18,333 scaffolds. These SNPs were consolidated by perfect linkage and then clustered at LOD scores of 6.1, 6.0, and 5.5. Combining scaffolds from the core linkage groups with those clustered among the full-sibling group ultimately provided a combined linkage set, with linkage group assignments to 16,285 scaffolds, representing 75.0 Mbp.

Mapping of scaffolds from just the core linkage groups to Coffea pseudochromosomes found several linkage group/pseudochromosome “best hit” pairs (e.g., most Asclepias scaffolds from a linkage group mapped to one pseudochromosome, while few scaffolds from other linkage groups mapped to that pseudochromosome). Asclepias linkage groups 2, 4, 6, 7, 8, and 9 mapped in this manner to Coffea pseudochromosomes 10, 8, 6, 11, 3, and 1, respectively (Figs. 4 and 5). From these six linkage groups, SNPs were chosen mapping to every one Mbp region (if available) of the corresponding Coffea pseudochromosome. Recombination distances were measured among these markers and their relative positions within Asclepias plotted against their position in Coffea (Figs. S2–S7). Monotonically increasing or decreasing series of points in these plots represent loci in Asclepias and Coffea that maintain their relative positions. Several such marker clusters are seen in these plots (Fig. S3), though they tend to cover only short chromosomal regions and are often interrupted by markers from outside the cluster.

Figure 4 Counts of Asclepias linkage group scaffolds mapping to Coffea pseudochromosomes.

Each column includes scaffolds from a single Asclepias linkage group, each row includes scaffolds mapping to a Coffea canephora pseudochromosome. Coffea chromosome 0 represents unassigned Coffea regions. Dot size is proportional to the number of mapping scaffolds, which is also provided.

Figure 5 Asclepias linkage group scaffolds mapped to Coffea pseudochromosomes.

Coffea canephora pseudochromosomes are shown in rows; the x-axis shows distance along each pseudochromosome. Each vertical bar represents one scaffold from the Asclepias core linkage groups, colored by its linkage group membership.

Progesterone 5β-reductase gene family

One region on linkage group 11 had 98.4% identity with peptide sequence from P5βR from Asclepias curassavica (Table S2). This region was supported by Asclepias syriaca transcriptome evidence, as well as mapped Calotropis transcripts and Coffea proteins. Approximately 500 bp downstream from this gene, a second region was identified sharing 52% amino acid identity with the first region, for 70% of its length. The second region lacks transcript evidence from Asclepias syriaca, though portions of Calotropis transcripts and Coffea peptides map to it. Gene predictions from Augustus and SNAP include potential exons within the region, and the region includes P5βR conserved motifs I, II, and III, and portions of motifs IV, V, and VI described by Thorn et al. (2008). It is interpreted here as a pseudogene of P5βR, ΨP5βR (Table S2).

Paralogs of P5βR have been described in other angiosperms including Arabidopsis, Populus, Vitis, and Digitalis, and the P5βR2 paralog occurs on a chromosome separate from that of P5βR1 in Arabidopsis and Populus (Pérez-Bermúdez et al., 2010; Bauer et al., 2010). Due to frame shifts and ambiguous exon boundaries in ΨP5βR, it is difficult to assess the correct peptide sequence it initially encoded, and therefore difficult to fully align with Digitalis P5βR1 and P5βR2 sequences. However, a few motifs, particularly a triple tryptophan at the N-terminal end of the sequence, suggest its origin from P5βR1, a conclusion supported by its position adjacent to the coding P5βR in Asclepias.

A third region on an unlinked scaffold exhibited moderate (37%) identity with the peptide sequence from linkage group 11 (Table S2). This region includes an intact reading frame and is matched by transcripts from Calotropis, though a lack of Asclepias transcripts matching this region indicates that it may not be regularly expressed within leaves or buds. A peptide alignment was made for this sequence, the known coding P5βR in Asclepias, and P5βR sequences from Asclepias curassavica, Calotropis procera, Calotropis gigantea, Rhazya, Digitalis, Catharanthus, and Picea to infer the phylogeny of this locus. The optimal model of sequence evolution selected by AIC was the LG+G+I model of peptide substitution, rate variation among sites, and proportion of invariable sites (BIC selected the LG+G model, but tree topologies were identical and are not shown). A maximum-likelihood estimate of the P5βR gene tree grouped the unlinked Asclepias sequence with a paralog from Rhazya (originating on supercontig 3 from Sabir et al., 2016) and Catharanthus paralog P5βR6 (Fig. 6). Together these are sister to all other P5βR sequences analyzed, except Picea, which was used to root the gene tree. The P5βR sequence from linkage group 11 is strongly supported as the most closely related sequence to the one from Asclepias curassavica, within a clade including P5βR1 sequences from Digitalis and Catharanthus.

Figure 6 Maximum likelihood phylogeny of progesterone 5β-reductase paralogs.

Asclepias syriaca labels indicate the linkage group from which that sequence originates. Catharanthus and Digitalis labels indicate numbered paralogs isolated from that species. Rhazya labels indicate the originating supercontig from Sabir et al. (2016) with two paralogs coming from supercontig 17. Calotropis procera labels indicate the originating transcript from Kwon et al. (2015). Calotropis gigantea labels indicate the originating contig from Hoopes et al. (2018). Numbers at nodes indicate aBayes support values. Branch lengths are in substitutions per site.

Analysis of the P5βR gene family across Apocynaceae showed that this gene family is largest in Rauvolfia, Catharanthus, and Tabernaemontana, with most of the expansion occurring in the common ancestor of these three (Fig. 3). However, this interpretation may change as more Apocynaceae genomes and transcriptomes become available.

Discussion

The Asclepias syriaca nuclear genome assembly presented here represents a large fraction of the protein-coding gene space, despite very high levels of heterozygosity and sequence data restricted to Illumina short reads. Gene space coverage is supported by high proportions of BUSCO plant core genes found within the assembly (93.6%) as well as assembled transcripts mapping to the assembly (96.7%). A substantial portion of genes from related plant species mapped to the assembly as well, including 88.4% of transcripts from Calotropis and 90.6% of amino acid sequences from Coffea.

Overall, the Asclepias assembly is fragmented when compared to other plant genomes assembled using either long reads or deep sequencing of known contiguous fragments (e.g., BACs or fosmids), and inclusion of these technologies in future assembly efforts should result in a more complete and contiguous assembly. Assembly was also hindered by poor quality mate-pair libraries containing low proportions of properly paired fragments (Table 2). However, assembly results are typical for a sequencing project relying entirely on short reads, especially for organisms with high levels of heterozygosity. For example, the Asclepias N50 value of 3.4 kbp compares favorably to the assembly of the rubber tree, Hevea brasiliensis, genome (N50 = 2,972 bp; Rahman et al., 2013), though it is not as contiguous as the dwarf birch, Betula nana, genome (N50 = 18.6 kbp; Wang et al., 2012), which incorporated several mate pair libraries. The assembly of the olive tree, Olea europaea, genome was also very similar to Asclepias, with N50 = 3.8 kbp prior to the inclusion of fosmid libraries (Cruz et al., 2016). However, paired-end and mate-pair data for Calotropis gigantea provide an example of a less fragmented assembly from similar data (N50 = 805 kbp, Table 1; Hoopes et al., 2018). The effect of high heterozygosity is clearly seen in the comparison of Asclepias and Catharanthus assemblies (Kellner et al., 2015). While sequence data and genome assembly methods are similar between the two, Asclepias has an estimated heterozygosity rate of >1 SNP per 20 bp, whereas the heterozygosity rate in the inbred Catharanthus cultivar is estimated at <1 SNP per 1,000 bp. This resulted in a N50 of 27.3 kbp assembled from only a single Catharanthus Illumina library (Table 1).

Functional annotations were applied to a high proportion (95.0%) of the 14,474 called genes, which were mapped to proteins from Catharanthus roseus and/or to Coffea canephora. The number of called genes is well below the typical value for plant genomes: the genome of Calotropis gigantea, the closest relative with an assembled genome, contains 19,536 gene loci (Hoopes et al., 2018). The genomes of Rhazya and Catharanthus contain 21,164 and 33,829 called genes, respectively (Kellner et al., 2015; Sabir et al., 2016). The genome of Coffea contains 25,574 protein-coding genes, and the genome of tomato, Solanum lycopersicum, from the sister order, Solanales, contains 36,148 (The Tomato Genome Consortium, 2012; Denoeud et al., 2014).

It is likely that the gene count in Catharanthus is an overestimate, a possibility in fragmented genome assemblies (Denton et al., 2014), as indicated by the excess of short predicted proteins relative to Coffea and Calotropis (Fig. 2). By contrast, the 14,474 called genes in Asclepias is likely an underestimate of the true number. While the size distribution of predicted Asclepias proteins is quite similar to that of Coffea, Asclepias contains fewer proteins of all sizes, and the dramatic reduction of orthogroups found in Asclepias relative to other species argues for deficiency in gene calling. While it’s possible that similar genes were mistakenly collapsed into a single contig during the assembly stage meant to collapse alleles at a single locus, this should only occur with genes isolated on small contigs and should not affect the number of orthogroups identified. Nevertheless, the high proportion of matches between the Asclepias genome assembly, Asclepias transcripts, and gene sets from related organisms, indicates that the assembly likely does contain sequence information for nearly the full complement of genes, but that some of these have not been recognized by gene calling algorithms due to the fragmented nature of the assembly.

Synteny within Gentianales

A total of 11 core linkage groups were produced from the set of SNPs originating from the maternal parent, matching the expected number from a single parent with x = n = 11 chromosomes. Using full-siblings, a set of SNPs originating from either the maternal or paternal parent clustered into hundreds of groups. A total of 22 of these were substantially larger than the others, matching the expectation of 22 linkage groups originating from two x = 11 parents.

Six of the 11 core linkage groups in Asclepias show high synteny at a chromosomal scale with the pseudochromosomes of Coffea (Figs. 4 and 5). This suggests that these chromosomes have remained largely stable and retained the same gene content for over 95 Myr, throughout the evolution of the Gentianales (Wikström et al., 2015). These stable chromosomes may have remained largely intact for a much longer period as well. The stable Coffea pseudochromosomes (1, 3, 6, 8, 10, and 11) retain largely the same content as inferred for ancestral core eudicot chromosomes, exhibiting little fractionation, even after an inferred genome triplication at the base of the eudicots, 117–125 Myr ago (Jiao & Paterson, 2014; see Figure 1B in Denoeud et al. (2014)).

Despite the conservation of gene content, gene order within stable chromosomes may be more labile. Plots of recombination distance among markers in Asclepias against physical distance in Coffea show several sets of markers in Coffea that retain their relative order in Asclepias, but are frequently interrupted by loci found elsewhere on the same Coffea pseudochromosome. For example, within Asclepias linkage group 2 there is a set of markers that retain their same relative ordering from positions three million to eight million on Coffea pseudochromosome 10 (Fig. S2). However, these markers in Asclepias are interrupted by markers mapping to positions closer to the origin on the same Coffea pseudochromosome as well as a marker mapping to the far end. The most conserved synteny is between Asclepias linkage group 8 and Coffea pseudochromosome 3, which show complete synteny except for an apparent transposition of markers at positions two million and seven million on Coffea pseudochromosome 3 (Fig. S3).

Contrasting the stability in gene content of six Coffea pseudochromosomes, pseudochromosome 2 is inferred to contain portions of at least five ancestral core eudicot chromosomes. This suggests significant fractionation in this chromosome since the eudicot triplication event (Denoeud et al., 2014). Even between Coffea and Asclepias, pseudochromosome 2 maps to portions of several Asclepias linkage groups (Figs. 4 and 5). Therefore, the fractionation within this chromosome appears to have either occurred only within the branch leading from the Gentianales ancestor to Coffea, or occurred earlier and then continued along the branch leading to Asclepias. If the latter is true, then a higher frequency of rearrangement may be a characteristic of this chromosome within the Gentianales, relative to other chromosomes. Analyses of chromosomal rearrangements in Rhazya (Figure 1 in Sabir et al. (2016)) support this view, suggesting several rearrangements between the core eudicot triplication event and the Gentianales ancestor, and continued rearrangement between that ancestor and Rhazya. However, mapped genomic resources within other Asterids outside of Gentianales are scarce, and are only found in taxa that have undergone additional genome duplication events since the eudicot triplication (e.g., Solanum, Daucus; The Tomato Genome Consortium, 2012; Iorizzo et al., 2016), complicating synteny assessments that might resolve the timing of fractionation of this chromosome.

The production of physical maps of both Asclepias and Coffea chromosomes will help resolve how frequently synteny has been disturbed between the two taxa. The ordered scaffold maps presented here (Figs. S2–S7) contain only a few dozen markers, and trends apparent now could be altered on maps with much greater resolution. The Coffea pseudochromosomes, meanwhile, are still ultimately ordered by recombination frequency, and about half of the scaffolds are placed with unknown orientation (Denoeud et al., 2014), which could manifest here as apparent transpositions among adjacent markers.

Progesterone 5β-reductase gene family

The name Asclepias comes from the Greek god of medicine, Asclepius, whose name was applied to this genus for its potent secondary compounds. The cardenolides of Asclepias belong to a class of steroidal compounds, cardiac glycosides, used to treat cardiac insufficiency. While the genetic pathway that produces β-cardenolides (the form of cardenolide that includes the medicinal compound digitoxin) is largely unknown, one of the early steps involves the conversion of progesterone to 5β-pregnane-3,20-dione (Gärtner, Wendroth & Seitz, 1990; Gärtner, Keilholz & Seitz, 1994), catalyzed by the enzyme P5βR. Orthologs of P5βR occur broadly across seed plants, even in taxa that do not produce β-cardenolides, including Asclepias, which only produces α-cardenolides (Bauer et al., 2010). The P5βR1 locus has been characterized in Asclepias curassavica, but information about its genomic context has remained unknown.

A coding P5βR ortholog was located in Asclepias syriaca on linkage group 11, sharing 98.4% amino acid identity with P5βR from Asclepias curassavica. This gene is supported by transcripts from Asclepias, as well as mapped transcripts from Calotropis and proteins from Coffea. The presence of a novel P5βR pseudogene was also identified closely downstream from the expressed gene (Table S2). Sharing high identity with the expressed P5βR, including several conserved motifs, it clearly originated from a P5βR duplication at some point. However, it is assumed to be non-functional due to its degraded exons interrupted by multiple stop codons and lack of expression evidence from the transcriptome.

A third region in Asclepias, on an unlinked scaffold, was matched by multiple P5βR sequences from Catharanthus (Table S2). This region is made up of a single ORF that shares only moderate identity with the Asclepias coding P5βR, and is not supported by Asclepias transcript evidence. In a P5βR phylogeny, the unlinked Asclepias region is sister to Catharanthus P5βR6 and a copy from Rhazya (Kellner et al., 2015; Sabir et al., 2016). These sequences together are sister to all other P5βR sequences analyzed except Picea, which was used for rooting (Fig. 6).

While at least two P5βR paralogs have been identified in a wide range of plants, and Rhazya, Rauvolfia, Catharanthus, and Tabernaemontana exhibit expression evidence of multiple paralogs, Asclepias is reduced for this group of genes. Rauvolfia and Tabernaemontana are known to produce cardenolides, but Catharanthus and Rhazya do not (Agrawal et al., 2012; Sivagnanam & Kumar, 2014; Abere et al., 2014; Hoopes et al., 2018). Calotropis is known to produce β-cardenolides (Bauer et al., 2010; Pandey et al., 2016), and contains two P5βR paralogs (Hoopes et al., 2018). It is possible that the fragmented nature of the current assembly precludes identification of all existing P5βR paralogs in Asclepias syriaca, however, both genome assembly and transcript evidence point toward one functional P5βR locus. While multiple genes are involved in the production of β-cardenolides, it may be that the reduction in the P5βR family is responsible for the lack of these compounds in Asclepias, which only contains α-cardenolides.

Conclusions

We present a draft genome assembly with linkage information of Asclepias syriaca, assigning nearly half of scaffolds to linkage groups. While the assembly remains fragmented, multiple lines of evidence indicate that nearly all of the gene space of Asclepias is represented within the assembly.

Linkage information allowed assessment of synteny across the order Gentianales. Six of 11 chromosomes retain similar gene content across the order, and these chromosomes have likely remained stable since the divergence of eudicots. One chromosome has either experienced dramatic fractionation since the divergence of Rubiaceae from other Gentianales, or experienced earlier fractionation that continued within Gentianales.

Asclepias syriaca and its relatives are important systems for a wide range of evolutionary and ecological studies, and are an important component of many ecosystems, serving as prolific nectar producers and as hosts to a range of specially adapted species. The availability of the Asclepias genome, coupled with genomic data from symbiotic organisms, particularly insects, promises to inform important mechanisms of co-evolution (Agrawal & Fishbein, 2008; Zhan et al., 2011; Edger et al., 2015). We expect that the data presented here will advance these studies and aid the discovery of novel insights into the origin and evolution of a charismatic family, the production of important secondary compounds, and the ecological and evolutionary relationships between milkweeds and their communities.

Supplemental Information

Supplemental Information 1 Supplementary figures and tables.

Click here for additional data file.

The authors wish to kindly thank the following for important contributions to this work: Winthrop Phippen for cultivating Asclepias, supplying tissue for sequencing, and harvesting fruits. Nicole Nasholm, Matt Parks, LaRinda Holland, Zoe Austin, and Lisa Garrison for DNA extraction and library preparation. The Oregon State University Center for Genome Research and Biocomputing for expert sequencing facilities and computational infrastructure. Access to the TAIR database was provided under the Terms of Use, accessed on August 11, 2016, available at http://www.arabidopsis.org/doc/about/tair_terms_of_use/417.

Additional Information and Declarations

Competing Interests

Author Contributions

DNA Deposition

Data Availability

The authors declare that they have no competing interests.

Kevin Weitemier conceived and designed the experiments, performed the experiments, analyzed the data, prepared figures and/or tables, authored or reviewed drafts of the paper, approved the final draft.

Shannon C.K. Straub conceived and designed the experiments, performed the experiments, analyzed the data, prepared figures and/or tables, authored or reviewed drafts of the paper, approved the final draft.

Mark Fishbein conceived and designed the experiments, performed the experiments, analyzed the data, contributed reagents/materials/analysis tools, authored or reviewed drafts of the paper, approved the final draft.

C. Donovan Bailey conceived and designed the experiments, performed the experiments, analyzed the data, authored or reviewed drafts of the paper, approved the final draft.

Richard C. Cronn conceived and designed the experiments, performed the experiments, analyzed the data, contributed reagents/materials/analysis tools, authored or reviewed drafts of the paper, approved the final draft.

Aaron Liston conceived and designed the experiments, performed the experiments, analyzed the data, contributed reagents/materials/analysis tools, authored or reviewed drafts of the paper, approved the final draft.

The following information was supplied regarding the deposition of DNA sequences:

The whole genome shotgun project and transcriptome shotgun assembly are available at DDBJ/ENA/GenBank: MSXX01000000 and GFXT01000000.

The following information was supplied regarding data availability:

A genome browser is available at www.milkweedgenome.org.

Additional data is available at the Oregon State University institutional archive: Weitemier, Kevin A. (2017): Supplemental data to “A draft genome and transcriptome of common milkweed (Asclepias syriaca) as resources for evolutionary, ecological, and molecular studies in milkweeds and Apocynaceae.” Oregon State University. Dataset. DOI 10.7267/N9M61HDR.

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
