# Peer review of "A draft genome and transcriptome of common milkweed (Asclepias syriaca) as resources for evolutionary, ecological, and molecular studies in milkweeds and Apocynaceae"

_PeerJ, doi:10.7717/peerj.7649_

## Round 0.1 · original submission · Minor Revisions

· Academic Editor

Minor Revisions

Your manuscript has been seen by two reviewers. Based on their detailed assessment and my own, I feel this manuscript is well suited for publication in PeerJ pending a few minor revisions.

Reviewer 1 ·

Basic reporting

The manuscript presented by the authors reports on a draft genome and transcriptome assembly of the common milkweed (Asclepias syriaca). They also discussed the structure of the genome in comparison to other related species, such as Coffea. I think this is a well performed analysis with clear implications as to why these data should be published and accessible. Overall, I found the paper to be very well written and the need for this work well articulated. I particularly appreciated the comparison of the Asclepias transcriptome with other Apocynacae transcriptomes.

Experimental design

The experimental design detailed by the authors was appropriate for this study. For the most part, the methods and statistical analyses were clear and a well-defined. I think the authors could be a bit more clear regarding the number of individuals sampled and sequenced, as it became a bit confusing at times in their sequencing details (see specific comments in the methods section below). However, in my opinion, this is something that can be easily addressed.

Validity of the findings

Based upon the methods used, the results of the study appear to be valid. The methods and analyses were appropriate. Their conclusions were well thought out and could be directly link to the results presented.

Additional comments

Specific comments:
Line 130: It would be better to give these in terms of final concentrations rather than number of microliters. Alternatively please indicate what total volume was used so that the audience can determine final concentration.

Line 141: “Frozen tissue was sent to GlobalBiologics..”
Is this from the same individual that was collected McDonough Co.?

Line 144: “Purified DNA was provided to the CGRB….”
Again, were all of the libraries of different sizes prepared for the same individual? I am assuming so, but I would just reiterate that because otherwise it is somewhat unclear.

Line 146: “… one of 15 samples pooled on a lane..”
I’m confused as to what these other 15 samples are from.

Line 148: ‘’ one of 3 samples pooled on a lane…’
See comment above.

RNA-seq methods: In my experience plants also contain a large number of ribosomal RNAs. Were these removed prior to library preparation?

Line 180-181: “…. To yield 101 bp single end reads…”
In my experience de novo transcriptome assembly is more efficient when using paired end reads. Why were single-end reads used?

Lines 323: I would suggest writing out Progesterone 5-beta reductase when first introducing it in the methods.

Line 387: Are the RNA oligos used to capture the SNPs listed somewhere?

Line 489: I’m not sure what the authors are referring to when they state…. “This represents a total of 865 BUSCO genes, or 93.6%.”

Line 483: Perhaps it would be a good idea to describe the transcriptome assembly in a bit more detail, such as providing N50 data, etc. I feel as though the information about the transcriptome gets a bit lost in this paragraph.

Lines 505-521: While interesting, it is unclear clear as to how gene death rates, birth rates, and innovation rates were calculated.

Line 580: Change ‘cosely’ to closely.

·

Basic reporting

The authors have presented nice, well written manuscript presenting a reference genome for Asclepias syriaca.

The introduction is well written overall, cited the relevant literature, and was concise. I felt that the first three paragraphs that were sometimes redundant in their structure and wording, particularly the listings of the multiple studies that use Asclepias as a model found at the end of each paragraph. A rewrite should address these issues.

In some places, I was confused about which set of reads were used for a particular analysis. An example being on line 392 where I wasn't sure which set of reads were used for the linkage analysis.

Images of Asclepias syriaca would be a great addition to this manuscript to showcase this beautiful and unique plant, particularly its unique floral structures. I suggest swapping images of the plant for Figure 1. I don’t think Figure 1 adds much to the main manuscript and might be better suited as a supplemental figure.

Minor points and other comments have been added under the General Comments section.

Experimental design

The materials and methods are largely well-written. There were several places where some revision would help with clarity and flow, particularly when describing how and where each library was constructed and sequenced. I particularly found the paragraph describing the mate-pair library (lines 141-151) construction to need revision. Other comments and questions have been included in the General Comments.

I was very pleased to see the section on contaminant removal. This is often an ignored, but very important step in genome projects.

Using only short read data to construct a reference genome is challenging and it appears that the authors undertook the appropriate methods for assembly and annotation.

If the authors are planning on future sequencing efforts to expand this genome or sequence additional Asclepias, it might make a nice addition to talk about these points in the Discussion.

Validity of the findings

I was initially concerned by the relatively large number of scaffolds and small N50 size, but I feel that the authors acknowledge this throughout. Their discussion of the challenges in assembling highly heterozygous genomes from only short read data and comparing this assembly to other similarly fragmented assemblies was a good inclusion to relieve these some of these concerns. Creating a reference genome for a non-model plant can be expensive given the large size of many plant genomes. It would be beneficial to eventually add long-read data to this project and examine what parts of the genome are missing from the current assembly.

Despite a very fragmented assembly, I found the linkage and synteny analyses quite interesting. The development of SNP markers using the assembly graph bubbles from Platanus was an approach I had not encountered before and seems to work as intended for creating a targeted enrichment probe set. On line 399, the authors state that they called SNPs using bcftools call. In my experience, SNPs are called using the mpileup and view commands. A clarification here about the approach and specific commands used would be helpful.
Analyses identified 11 linkage groups from the full set of individuals and 22 groups from the full-sib set. Asclepias and Coffee both have a chromosome count of n=11, 2n=22. It would be very interesting for the authors to expand more on the connection between 11 linkage groups and 11 chromosomes in their discussion.

Overall, I found the Discussion to be well-written and thoughtful.

Additional comments

Some minor points and areas for clarification:

L131: What were the settings for sonication on the BioRuptor and what was the desired fragment size?

L132: What was the library kit used to prepare these two Illumina libraries?

L153: Were the raw read files assessed for quality, overrepresented sequences, etc. using FastQC or similar tool?

L156-157: Please specify what commands/parameters were specified for Trimmomatic, including which set of Illumina adapter sequences were used for search and removal.

L176: Was the RNA checked for quality and integrity prior to library preparation? If so, please specify here.

L182: Similar to above, please specify the commands used for Trimmomatic.

L186-188: It was mentioned on line 178 that strand-specific RNA-seq libraries were prepared and I see no mention of any read orientation specified to Trinity (--SS_lib_type). If you didn’t provide read orientation, Trinity treats the reads as non-strand specific. I’m curious if you re-ran your Trinity assembly specifying the proper read orientation for strand-specific libraries using the dUTP method (--SS_lib_type RF) if you would find any differences or additional transcripts.

L192-193: Which databases did Mercator and TRAPID search against for functional annotation? The choice of database, such as the NCBI Nr or UniProt, provide different levels of information. Usually doing both, with higher priority given to the curated UniProt database provides enough information to confidently assign putative function. Additionally, were any specific options used for Mercator or TRAPID, or the default settings? It would be good to specify.

L198-200: Please provide the NCBI accession numbers here for the downloaded Rhayza and Calotropis transcriptomes.

L286 and L483: What do you mean by “best” ORF?

L468: I am curious to know how well the scaffolding performed. How many contigs were initially assembled and how many were able to be scaffolded using your mate-pair information? I think that would be an interesting piece to add here to demonstrate how well your final scaffolds look.

---

## Round 0.2 · Minor Revisions

· Academic Editor

Minor Revisions

Your revised manuscript has addressed the previous concerns raised by the reviewers, but there are still a few minor data curation and availability issues that were raised by the Section Editor. The authors should carefully consider each of these points and provide links to the requested data where possible:

1. The manuscript title presents a draft genome without actually making one present. There were over 50000 scaffolds assembled and aligned to a near-relative reference genome, but the actual data representing those alignments are not readily available. In Table 2 information points to the raw data, but a resolvable scaffolds would not be available without another resource.

2. Further descriptions focus on key genes of interest; however, the sequences for these, or the context for these sequences is not available. There is discussion about performing gene annotation and even detecting SNPs, yet none of this data is displayed or available. The readers need to be pointed to the resources for scaffold files, gene lists, and candidate mapping locations as it seems there was an assembly with no scaffolds presented, gene prediction and annotation with no lists presented, linkage mapping with nothing to be linked except visualized, SNPs without descriptions. Likewise, for a gene list it would be helpful to see the annotations go to the extent of providing an accepted annotation methodology, such as with the gene ontology GO: annotations.

3. As another note: Journal manuscripts are often scanned by text-mining software that locates and extracts core data elements, like gene function. Adding standard ontology terms, such as the Gene Ontology (GO, geneontology.org) or others from the OBO foundry (obofoundry.org) can enhance the recognition of your contribution and description. This will also make human curation of literature easier and more accurate.

---

## Round 0.3 · accepted · Accept

· Academic Editor

Accept

Thank you for updating the data availability section of the manuscript and for releasing additional genomic resources and functional annotations for milkweed. These additional resources will be useful to the broader comparative genomics community and this manuscript is now suitable for publication.